# Femtosecond Laser Direct Writing of Integrated Photonic Quantum Chips for Generating Path-Encoded Bell States

**DOI:** 10.3390/mi11121111

**Published:** 2020-12-15

**Authors:** Meng Li, Qian Zhang, Yang Chen, Xifeng Ren, Qihuang Gong, Yan Li

**Affiliations:** 1State Key Laboratory for Mesoscopic Physics and Frontiers Science Center for Nano-Optoelectronics, School of Physics, Peking University, Beijing 100871, China; mengli2016@pku.edu.cn (M.L.); zhangqianlucy@126.com (Q.Z.); qhgong@pku.edu.cn (Q.G.); 2CAS Key Laboratory of Quantum Information, University of Science and Technology of China, Hefei 230026, China; cy123@mail.ustc.edu.cn (Y.C.); renxf@ustc.edu.cn (X.R.); 3Collaborative Innovation Center of Extreme Optics, Shanxi University, Taiyuan 030006, China; 4Peking University Yangtze Delta Institute of Optoelectronics, Nantong 226010, China

**Keywords:** photonic quantum chip, femtosecond laser direct writing, Hadamard gate, CNOT gate, path-encoded Bell state

## Abstract

Integrated photonic quantum chip provides a promising platform to perform quantum computation, quantum simulation, quantum metrology and quantum communication. Femtosecond laser direct writing (FLDW) is a potential technique to fabricate various integrated photonic quantum chips in glass. Several quantum logic gates fabricated by FLDW have been reported, such as polarization and path encoded quantum controlled-NOT (CNOT) gates. By combining several single qubit gates and two qubit gates, the quantum circuit can realize different functions, such as generating quantum entangled states and performing quantum computation algorithms. Here we demonstrate the FLDW of integrated photonic quantum chips composed of one Hadamard gate and one CNOT gate for generating all four path-encoded Bell states. The experimental results show that the average fidelity of the reconstructed truth table reaches as high as 98.8 ± 0.3%. Our work is of great importance to be widely applied in many quantum circuits, therefore this technique would offer great potential to fabricate more complex circuits to realize more advanced functions.

## 1. Introduction

In recent years, integrated photonic quantum chips have become a hot topic in quantum optics field, for its scalability, stability, and miniaturization compared with bulk optics. They are fabricated by the silicon-based lithography [1,2,3,4,5,6], the femtosecond laser direct writing (FLDW) [7,8,9], and a new emerging platform based on lithium niobate on insulator (LNOI) [10,11,12]. Silicon-based waveguide photonic chips have maturely developed using silica on silicon [1], silicon on insulator [4], silicon nitride [5], silicon oxynitride [6] and so on, while their waveguides can only support single polarization mode due to the rectangular cross-section, and the conventional lithography is limited to the planar layout. The FLDW can realize 3D fabrication of waveguides with near round cross-section [13,14,15]. Therefore, it can realize quantum information processing not only by path encoding [8,16] but also by polarization encoding [17,18]. By virtue of its true three-dimensional direct writing ability, FLDW can fabricate more flexible and complex quantum circuits with 3D structures, simplify the layout and reduce the number of elements required [19,20,21]. Nowadays, photonic quantum chips realized by FLDW technique have been widely applied in various research, such as quantum logic gates [8,17], quantum algorithm [22,23,24], quantum walk [25,26,27], quantum simulation [28,29,30], quantum key distribution [31,32], boson sampling [33,34], entangled photon sources [35], and so on.

Quantum logic gates are the basic elements of quantum circuits. Combining several simple logic gates together can construct more complex logic gates to perform quantum algorithms and realize specific functions [36,37,38,39]. The Hadamard (H) gate and the Controlled-NOT (CNOT) gate are the most basic and important single and two qubit gates, respectively. Cascading the H gate and the CNOT gate together can generate Bell states, which are also the most basic entangled states and serve as a central physical resource in various quantum information protocols like quantum cryptography, quantum teleportation, entanglement swapping, and in tests aimed at excluding hidden variable models of quantum mechanics [40]. Up to now, individual path-encoded or polarization-encoded H gate or CNOT gate on chips has already been fabricated by FLDW technique [16,17,18,41]. However, the combination of one H gate and one CNOT gate in a single photonic chip for generating path-encoded Bell states has not been reported. Here we have successfully fabricated an integrated photonic quantum chip composed of H and CNOT gates by FLDW technique to generate all four path-encoded Bell entangled states, whose average fidelity is 98.8 ± 0.3%, which is higher than that of a silicon-based photonic chip (~91.2 ± 0.2%) [1]. This combination is very useful in many quantum circuits, especially in quantum algorithms circuits [2,19,42]. The capacity of FLDW to fabricate such a chip with high quality and fidelity provides a possibility for the fabrication of large-scale 3D functional integrated photonic quantum chips.

## 2. Design of the Photonic Quantum Chip

Figure 1a shows the schematic configuration of the photonic quantum chip composed of one H gate and one CNOT gate. The H gate (red dashed box) is a balanced directional coupler (DC), and the CNOT gate (the rest of the whole chip) is based on our previous work [16] but with improved symmetry in circuit design. The left represents the input ports (0–5) and the right the output ports (0′–5′). The power reflectivity $R=P_{\mathrm{OUT}1}/(P_{\mathrm{OUT}1}+P_{\mathrm{OUT}2})$ is marked on each DC, defined by the ratio of the output power from OUT1 to the total output power of a DC where the laser is launched into IN1 as shown in Figure 1b. The symmetry of the DC guarantees that the same relations hold when light is launched into port IN2, by simply inverting the two indices. There are three DCs with reflectivity of 1/2 and three DCs with reflectivity of 1/3. The control qubit $C_{q}$ (target qubit $T_{q}$) is encoded via spatial paths $C_{0}$($T_{0}$) and $C_{1}$($T_{1}$). The remaining two paths are ancillary vacuum modes to complete the network. The H gate in this circuit represented by a DC is indeed a Hadamard-like gate $H^{\prime }=e^{i\pi /2}e^{-i\pi \sigma_{z}/4}He^{-i\pi \sigma_{z}/4}=\frac{1}{\sqrt{2}}\left(\begin{array}{cc} 1 & i\\ i & 1 \end{array}\right)$ as shown in Reference [3], where $H=\frac{1}{\sqrt{2}}\left(\begin{array}{cc} 1 & 1\\ 1 & -1 \end{array}\right)$ is the standard Hadamard gate. They are equivalent up to local $\sigma_{z}$ rotations, so we still use H gate for description in this article. When a single photon is input to path $C_{0}$($C_{1}$) representing the logic state ${|0\rangle }_{c}$(${|1\rangle }_{c}$), the state will be unitarily transformed by the H gate to generate a superposition state $\frac{1}{\sqrt{2}}\left({|0\rangle }_{c}+i{|1\rangle }_{c}\right)$ ($\frac{1}{\sqrt{2}}i\left({|0\rangle }_{c}-i{|1\rangle }_{c}\right)$). The rest of the chip is a path encoded probabilistic CNOT gate. In the CNOT gate, when the control qubit $C_{q}$ is in logic state $|1\rangle$, the target qubit $T_{q}$ will flip from initial logic state $|0\rangle$($|1\rangle$) to opposite logic state $|1\rangle$($|0\rangle$). However, when $C_{q}$ is in logic state $|0\rangle$, the logic state of $T_{q}$ remain unchanged. When one photon from $C_{q}$ and the other photon from $T_{q}$ are mixed in the central DC with reflectivity of 1/3 simultaneously, they will undergo a partial bunching effect and get a π phase shift due to Hong–Ou–Mandel (HOM) interference of indistinguishable photons [16,43]; and this π phase shift will change the output state of Mach–Zehnder (MZ) interferometer connected by paths $T_{0}-T_{1}^{\prime }$ and $T_{1}-T_{0}^{\prime }$.

Now, when the control qubit $C_{q}$ is in superposition state of $|0\rangle$ and $|1\rangle$, the chip will generate all four path-encoded Bell states with the entangling function of the CNOT gate. The relationships are as follows:(1)$$\begin{array}{l} {|00\rangle }_{ct}\to \frac{1}{\sqrt{2}}i\left({|00\rangle }_{ct}-{|11\rangle }_{ct}\right),\\ {|01\rangle }_{ct}\to \frac{1}{\sqrt{2}}i\left({|01\rangle }_{ct}+{|10\rangle }_{ct}\right),\\ {|10\rangle }_{ct}\to -\frac{1}{\sqrt{2}}\left({|00\rangle }_{ct}+{|11\rangle }_{ct}\right),\\ {|11\rangle }_{ct}\to -\frac{1}{\sqrt{2}}\left({|01\rangle }_{ct}-{|10\rangle }_{ct}\right). \end{array}$$

Compared with the standard quantum circuit composed of one H gate and one CNOT gate, the output Bell states for input state ${|00\rangle }_{ct}$ and ${|10\rangle }_{ct}$ in Equation (1) should be exchanged, because the H gate here is indeed an H′ gate.

## 3. Experimental and Results

The photonic quantum chip is directly written inside borosilicate glass (Eagle2000, Corning) by focusing the femtosecond laser pulses produced by a regeneratively amplified Yb: KGW femtosecond laser system (Pharos-20W-1MHz, Light Conversion). The 1030 nm laser pulses with duration of 240 fs at repetition rate of 1 MHz are focused 170 μm beneath the surface of the glass by a microscope objective with a numerical aperture (NA) of 0.5 (RMS20X-PF, Olympus). The sample is translated at constant speed executed by a computer-controlled high-precision three-axis air-bearing stage (FG1000-150-5-25-LN, Aerotech).

The first step is to fabricate straight waveguide to determine the optimal parameters by scanning pulse energy and translation speed. In this work, they are 386 nJ and 20 mm/s. Figure 2a shows the optical micrograph of the cross section of the fabricated straight waveguide, with a size of 4.5 × 7.1 μm. The mode distributions of the 785 nm laser guided in the fiber and in the waveguide with two orthogonal polarizations are shown in Figure 2b–e. The mode field diameter (MFD) of the guided mode in the waveguide is 6.1 × 6.6 μm (5.9 × 6.6 μm) in H (V) polarization with an almost round-shaped mode field. The mode in the waveguide slightly deviates from that in the fiber, which results in coupling loss due to the mode mismatching. The measured insertion loss of the 2.5 cm straight waveguide is 2.52 dB (2.43 dB), and the coupling loss is 0.80 dB (0.73 dB), and the Fresnel reflection loss is 0.177 dB/facet, so the propagation loss is 0.55 dB/cm (0.54 dB/cm) for H (V) polarization, which is better than 0.7 dB/cm in our previous work [16].

Using the same parameters, we fabricated a series of directional couplers with different interaction lengths L at fixed interaction distance d. For the curved segments of DC, the bending radius is set as 60 mm to guarantee a low bending loss. The spacing between two input (output) ports of the DC is 127 μm to match the pitch of the fiber array. The interaction distance is set as d=8 μm to acquire a high coupling coefficient κ but without waveguide-overlapping geometrically, according to previous experimental results [16]. We change the interaction lengths in a wide range from 0–5.5 mm to find the optimal parameters. As shown in Figure 3a, the fitting curves of the measured reflectivity *R* and transmission *T* of DCs varying with interaction length *L* follow the curves in the forms of $\cos^{2}\varphi$ and $\sin^{2}\varphi$ very well, respectively. Therefore, we can get the linear relation between the coupling phase φ and the interaction length L: $\varphi =\kappa L+\varphi_{0}$, which is shown in Figure 3b, so that we can conveniently estimate the *L* for *R* = 1/3 from the data for *R* = 1/2. To fabricate DCs with *R* = 1/2 and *R* = 1/3, we only focus on the range of L from 0–1 mm. According to the design of the circuit in Figure 1a, we fabricated several chips composed of one H gate and one CNOT gate with slightly different interaction lengths *L* around the values of $L_{1/2}$ and $L_{1/3}$ to take into account possible fabrication imperfections. Finally, we successfully find satisfactory interaction lengths for DCs with *R* = 1/2 and *R* = 1/3, which are $L_{1/2}=$ 0.450 mm and $L_{1/3}$ = 0.716 mm, respectively, and the chip size is about 635 μm × 2.5 cm.

A classical characterization of the quantum chip is performed by injecting 808 nm CW laser beam in V polarization into the chip to measure the output power of each port, we select out the best one whose results are close to the theoretical values to perform quantum characterization. In classical characterization, we input CW laser into each input port of the chip and use the power meter to record the ratio of the output power for each output port. The theoretical prediction of the ratio of output power for each input case of the circuit is listed in Table 1, and the experimental classical characterization results are shown in Table 2, where $F=\sum_{i}\sqrt{p_{i}q_{i}}$ ($p_{i}$ is the theoretical value, and $q_{i}$ is the experimental one) represents the fidelity of each row. In view of the fabrication imperfection and slightly asymmetric beam splitting ratio of fabricated DCs for different input ports, the experimental results deviate a little from the theoretical values, but they are acceptable according to the calculated fidelity of each row. Moreover, the latter quantum characterization experimental results also confirm the high quality of this chip.

For the quantum characterization of the selected chip, we inject the time-correlated photon pairs into the chip and measure the coincidence counts of output photons. The experimental setup of the quantum characterization system is shown in Figure 4. The 808 nm dual photon pairs were generated by pumping a 0.5 mm thick beta-barium borate (BBO) crystal using 140 mW, 404 nm CW laser (ECL801, Uni Quanta) through Type-I spontaneous parametric down-conversion (SPDC) process. The photon pairs are divided into two parts and deflected by small prisms. After passing through long pass filter (LPF), half-wave plate (HWP), interference filter (IF), photons are collected by coupling lens into single mode fibers (SMFs). The LPF from 650 nm is used to remove the scattering 404 nm pump light, and the IF with 3 nm bandwidth is used to ensure good spectral indistinguishability. The HWP and the fiber polarization controller (PC) can control the polarization state of photon in fiber to maintain V polarization. One way is inserted into a delay line to control the relative arrival time of two photons to ensure the temporal indistinguishability. The photons are injected into the chip through 4-channel input fiber array (FA) with the same 127 μm spacing and then collected from the chip to the output fiber array. After that, the output photons are detected by single photon counting modules (SPCMs) (Excelitas, SPCM-800-14-FC) and conveyed to the Time to Digital Converter (TDC) (ID800, IDQ) to conduct the coincidence counting of the output photon pairs.

By scanning the relative delay between two input photons, we can get coincidence counting curves for four kinds of output-photon combinations after post-selection. As shown in Figure 5a, when the input-photon combination is (1,3), the interference curve of output-photon combination (2′,4′) shows a HOM dip at interference point, where the relative delay is zero and the corresponding coincidence counts reduce to zero. Its HOM interference visibility is about 98.5 ± 1.2% with accidental coincidence counts subtracted. However, the interference curves of output-photon combinations (2′,3′) and (1′,4′) slightly change, and their coincidence counts at interference point get close to each other. In their interference curves, the occurrence of small dip or peak is due to the deviation of reflectivity for DCs and difference of internal phase between two arms of the MZ interferometer in the fabricated photonic circuit. The coincidence counts of the remaining output-photon combination (1′,3′) are close to zero. This represents that the post-selected output-photon state is a path-encoded entangled state:(2)$${|00\rangle }_{ct}\Leftrightarrow {|13\rangle }_{ct}\to \frac{1}{\sqrt{2}}\left({|1^{\prime }4^{\prime }\rangle }_{c^{\prime }t^{\prime }}+e^{i\phi }{|2^{\prime }3^{\prime }\rangle }_{c^{\prime }t^{\prime }}\right)\Leftrightarrow \frac{1}{\sqrt{2}}\left({|00\rangle }_{c^{\prime }t^{\prime }}+e^{i\phi }{|11\rangle }_{c^{\prime }t^{\prime }}\right).$$

According to the theoretical prediction, the phase factor ϕ should be π, but we cannot determine its value directly in the chip. To completely analyze the output-photon state by acquiring the value of ϕ, we need to place more elements such as DCs and phase shifters behind the logic gates on the current chip [3], which need to lengthen the chip and fabricate the electrode or convert the encoding information of path to polarization outside the chip [35,44,45,46], which needs more optical elements and equipment. In Reference [1], the authors cannot determine the value of ϕ, but the demonstration of excellent logical basis operation of the CNOT gate and coherent quantum operation gives them great confidence. Similarly, as shown in Figure 5b–d, when the input-photon combinations are (1,4), (2,3), and (2,4), the visibilities of each interference curve for output-photon combinations (2′,3′), (2′,4′), and (2′,3′) are 97.8 ± 1.8%, 98.2 ± 1.8%, and 99.5 ± 0.5% with accidental coincidence counts subtracted, respectively. We can get three more path-encoded entangled state:
(3)$${|01\rangle }_{ct}\Leftrightarrow {|14\rangle }_{ct}\to \frac{1}{\sqrt{2}}\left({|1^{\prime }3^{\prime }\rangle }_{c^{\prime }t^{\prime }}+e^{i\phi^{\prime }}{|2^{\prime }4^{\prime }\rangle }_{c^{\prime }t^{\prime }}\right)\Leftrightarrow \frac{1}{\sqrt{2}}\left({|01\rangle }_{c^{\prime }t^{\prime }}+e^{i\phi^{\prime }}{|10\rangle }_{c^{\prime }t^{\prime }}\right),\;{|10\rangle }_{ct}\Leftrightarrow {|23\rangle }_{ct}\to \frac{1}{\sqrt{2}}\left({|1^{\prime }4^{\prime }\rangle }_{c^{\prime }t^{\prime }}+e^{i\phi^{\prime \prime }}{|2^{\prime }3^{\prime }\rangle }_{c^{\prime }t^{\prime }}\right)\Leftrightarrow \frac{1}{\sqrt{2}}\left({|00\rangle }_{c^{\prime }t^{\prime }}+e^{i\phi^{\prime \prime }}{|11\rangle }_{c^{\prime }t^{\prime }}\right),\;{|11\rangle }_{ct}\Leftrightarrow {|24\rangle }_{ct}\to \frac{1}{\sqrt{2}}\left({|1^{\prime }3^{\prime }\rangle }_{c^{\prime }t^{\prime }}+e^{i\phi^{\prime \prime \prime }}{|2^{\prime }4^{\prime }\rangle }_{c^{\prime }t^{\prime }}\right)\Leftrightarrow \frac{1}{\sqrt{2}}\left({|01\rangle }_{c^{\prime }t^{\prime }}+e^{i\phi^{\prime \prime \prime }}{|10\rangle }_{c^{\prime }t^{\prime }}\right).$$

In theory, these phase factors should be $\phi^{\prime }=0$, $\phi^{\prime \prime }=0$ and $\phi^{\prime \prime \prime }=\pi$, but these phases cannot be confirmed directly either. In the future, we will make efforts to improve our experimental methods to conduct a complete characterization of the generated entangled states. In addition to the interference curves for different input-output combinations, we also need to reconstruct the truth table to determine the fidelity of this chip.

Compared with our previous work of path encoded CNOT gate [16], the success of this work depends not only on the high interference visibility but also on the equal probability distribution of two terms in the entangled state (Equations (2) and (3)), which requires tougher fabrication. For each input-photon combination, we normalize the coincidence counts for each output-photon combination with accidental counts subtracted to calculate their corresponding probability. As shown in Figure 6, the probabilities of each computational-basis output for each computational-basis input are represented by the height of the filled pink bars, and the height of empty bars stands for the theoretical value. The reconstructed truth table of the chip coincides very well with the theoretical one and the average fidelity of the fabricated chip is as high as 98.8 ± 0.3%, which is also higher than 91.2 ± 0.2% in reference [1].

## 4. Discussion

The experimental results demonstrate that we can fabricate a photonic quantum chip constructed by cascading one H gate and one CNOT gate with high fidelity by FLDW technique. The H gate aims to prepare a superposition state of $|0\rangle$ and $|1\rangle$, and the post-selected CNOT gate generates a canonical two-qubit entangling gate. Therefore, this chip can generate all four path-encoded Bell entangled states. According to the classical characterization results, the performance of the fabricated chip can still be improved by more precisely controlled interaction lengths and a more stable micromachining system, but the improvement of fidelity is limited. The fabrication and measurement of this chip are more difficult than those of the CNOT gate, because the output probabilities of the two terms in the entangled state should be as equal as possible, which means that we need to make more efforts to control the fabrication details, adjust the alignment and coupling between fiber arrays and the chip, maintain the polarization of input photons and monitor the real-time detection process. Eventually, we successfully fabricated such a chip with a high fidelity of 98.8 ± 0.3%. This work is a demonstration of the cascading of one H gate and one CNOT gate. Furthermore, we can construct a quantum circuit to cascade and parallel several logic gates to generate more complex entangled states, such as GHZ state, but it requires more photons with quantum correlation [47,48], which is more difficult in measurement than in fabrication.

## 5. Conclusions

We realize the femtosecond laser direct writing of an integrated photonic quantum chip composed of one H gate and one CNOT gate for generating all four path-encoded Bell entangled states. For both classical and quantum characterization, the chip achieves great performance. The HOM interference visibilities of each input-output combination are all higher than 97.8 ± 1.8%, and the average fidelity of the reconstructed truth table is about 98.8 ± 0.3%, which suggest the high-quality of the fabricated chip. This basic quantum circuit is an important element and can be applied in many quantum computation algorithms, such as quantum Prime Factorization, quantum Grover Search and quantum Fourier Transform. Bell state is also an important entangled photon source, which is widely used in quantum communication and quantum computation. This work presents the capability of FLDW technique to fabricate more complex and functional photonic quantum computation chips.

## Figures and Tables

**Figure 1 micromachines-11-01111-f001:**
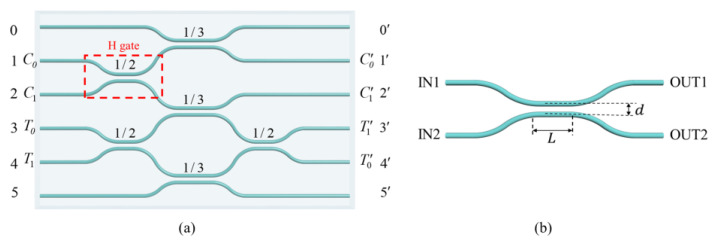
Schematic of the photonic quantum chip and the directional coupler (DC). (**a**) Schematic representation of a photonic chip composed of one H and one CNOT gate to generate path-encoded Bell states. The DC in the red dashed box represents an H gate. (**b**) Schematic representation of a waveguide directional coupler; IN1 and IN2 are the input ports, whereas OUT1 and OUT2 are the output ports of the device; *L* is the interaction length and *d* is the interaction distance in the coupling region of two adjacent waveguides.

**Figure 2 micromachines-11-01111-f002:**
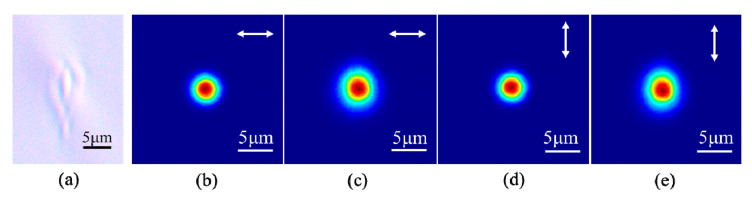
Optical micrograph and mode distributions of a fabricated straight waveguide. (**a**) Micrograph of the cross section of the waveguide. (**b**) Mode of the fiber in H polarization; (**c**) mode of the waveguide in H polarization; (**d**) mode of the fiber in V polarization; (**e**) mode of the waveguide in V polarization. The wavelength of the injected CW laser is 785 nm.

**Figure 3 micromachines-11-01111-f003:**
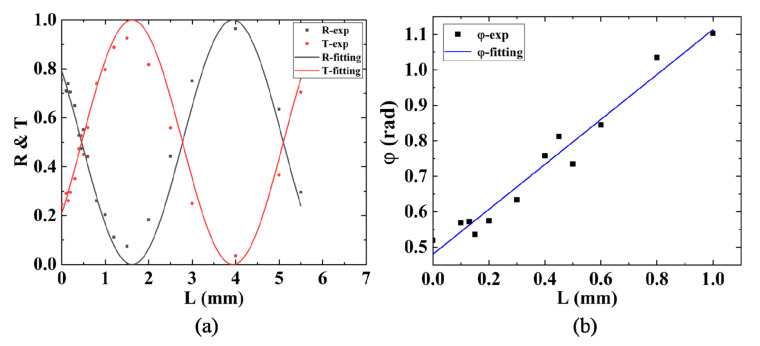
Experimental reflectivity and transmission of fabricated DCs. (**a**) Measured reflectivity *R* and transmission *T* of the DCs with different interaction length *L* at fixed interaction distance *d* = 8 μm as well as fitting curves for *R* and *T*. (**b**) Fitting for linear relation between coupling phase φ and interaction length *L* ranging from 0–1 mm.

**Figure 4 micromachines-11-01111-f004:**
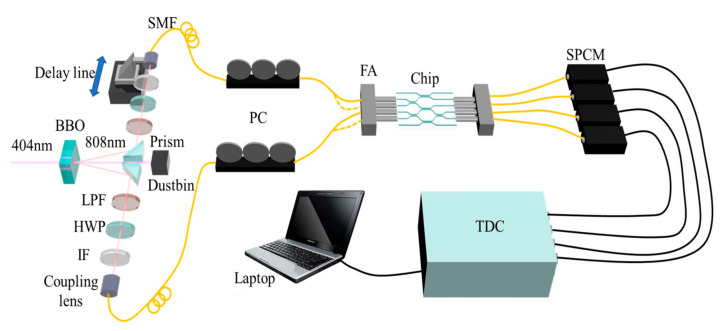
Experimental setup for quantum characterization. Through Type-I spontaneous parametric down-conversion (SPDC) process, 808 nm photon pairs are generated by pumping the BBO crystal using 140 mW, 404 nm CW diode laser. Long pass filter (LPF) from 650 nm and interference filter (IF) at 808 nm with 3 nm bandwidth are used to ensure spectral indistinguishability. Half-wave plate (HWP) and polarization controller (PC) are used to control the polarization state of photons in fiber. A delay line is inserted into one way to control the relative arrival time of photons to ensure temporal indistinguishability. Photons are injected into waveguides in the chip through fiber array and then collected at the output by another fiber array. Single photon counting modules (SPCMs) and the Time to Digital Converter (TDC) are used to conduct coincidence counting of different output-photon combinations.

**Figure 5 micromachines-11-01111-f005:**
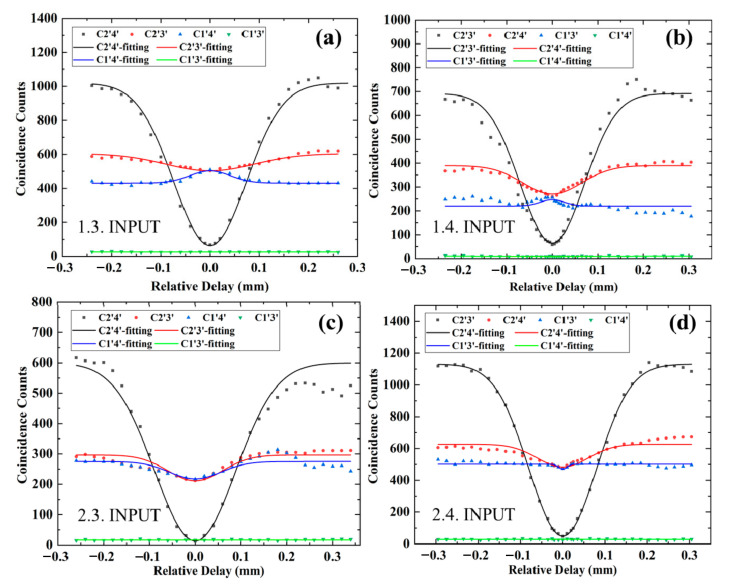
Coincidence counts in 30 s of post-selected output-photon combinations x′y′ denoted as Cx′y′ for different input-photon combinations xy denoted as x.y. INPUT as a function of the relative delay of photons input in x and y ports: (**a**) input (1,3); (**b**) input (1,4); (**c**) input (2,3); (**d**) input (2,4). From the interference curve with a deep dip, the HOM interference visibilities are (**a**) 98.5 ± 1.2%, (**b**) 97.8 ± 1.8%, (**c**) 98.2 ± 1.8%, (**d**) 99.5 ± 0.5%, respectively.

**Figure 6 micromachines-11-01111-f006:**
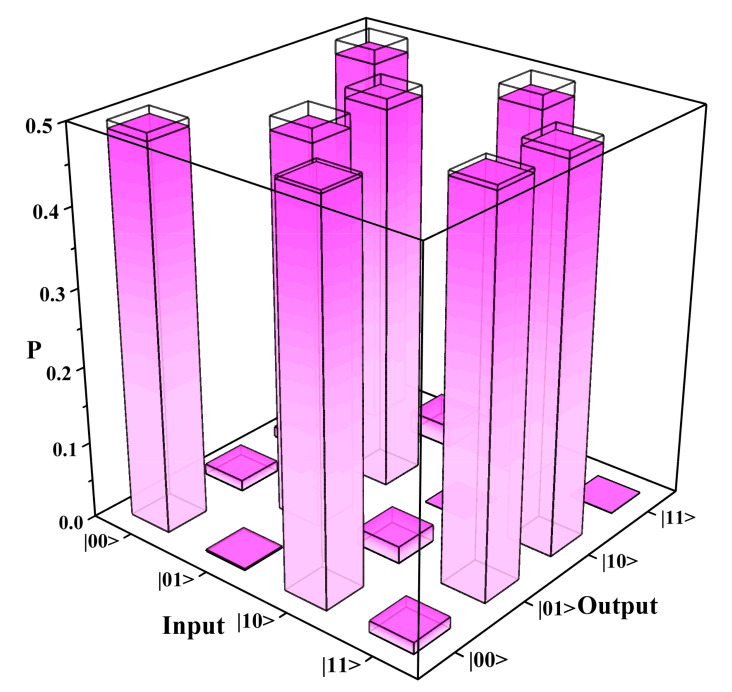
Reconstructed truth table of combined chip composed of one H gate and one CNOT gate. The labels on the Input axis represent $|C_{q}T_{q}\rangle$, and the labels on the Output axis represent $|C_{q}^{\prime }T_{q}^{\prime }\rangle$. *P* represents the probability for each input-output combination. The empty bars stand for the theoretical values, and the filled pink bars represent the experimental data. The average fidelity is as high as 98.8 ± 0.3%.

**Table 1 micromachines-11-01111-t001:** Theoretical prediction of the ratio of output power for each output port y′(0′–5′) when laser is injected into each input port x (0–5).

Theory	0′	1′	2′	3′	4′	5′
0	1/3	2/3				
1	1/3	1/6	1/6	1/6	1/6	
2	1/3	1/6	1/6	1/6	1/6	
3			1/3	0	1/3	1/3
4			1/3	1/3	0	1/3
5				1/3	1/3	1/3

**Table 2 micromachines-11-01111-t002:** Experimental classical characterization results of the selected chip for each input case.

Experiment	0′	1′	2′	3′	4′	5′	*F*
0	0.350	0.650					0.999
1	0.352	0.181	0.159	0.156	0.153		0.999
2	0.355	0.180	0.155	0.155	0.155		0.999
3			0.297	0.030	0.318	0.355	0.984
4			0.301	0.306	0.028	0.365	0.985
5				0.348	0.375	0.277	0.998

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
