# Peer review of "Femtosecond Laser Direct Writing of Integrated Photonic Quantum Chips for Generating Path-Encoded Bell States"

_micromachines, 2020, doi:10.3390/mi11121111_

Round 1

Reviewer 1 Report

This paper reports the application of femtosecond laser direct writing for generating integrated photonic quantum chips. The authors achieved basic functionalities with Hadamard and Controlled-NOT gates. Overall this works is well-presented, and it has demonstrated the potential of laser direct writing for the development of photonic quantum chips. I have no complaints on this paper.

Author Response

Thank you for your positive comments.

Reviewer 2 Report

The authors report on the ultrafast laser inscription of an integrated device for the on-chip generation of entangled path-encoded qubits pairs. The experiment is described in a complete way, the information provided are sufficient to understand all aspects and the experimental methods are clear. The demonstration of entanglement is only partial, because no quantum correlations have been actually analysed, but the authors recognize this limitation in the text and modulate their claims consequently. However, my main concern is related to the novelty of this experiment, which is somehow lacking. Integrated two-qubits logic gates for path engoding have been demonstrated extensively during the past decade by various research groups in the world, and the photon source used is a standard SPDC source. As so, there is nothing really new in this paper. Nevertheless, entanglement is a very useful resource for integrated quantum photonics technologies in general. As so, I believe that this manuscript deserves publication on Micromachines.

I have a minor comment for the revision of the text. In the introduction, the authors make a strong claim in saying that quantum photonic circuits are essentially made by either silicon-based or laser writing technologies.This is not true, as other platforms are relevant, e.g. Lithium Niobate technologies. In addition, The authors refer to silicon-based circuits mentioning only Silica-On-Silicon platforms (refs 1-3). This is a very skewed picture of the current state of the art in the field, as Silicon-On-Insulator also plays a major role, as well as Silicon Nitride/Oxinitride waveguides. Therefore, I suggest the authors to detail better this sentence.

Author Response

Thank you for your comments and suggestions.

  1. Using silicon-based photonic quantum chips, the generation of path-encoded Bell states by the same or similar structure as that in our work has been demonstrated during the past decade by various research groups to complete Bell-state analysis, state tomography and perform the teleportation protocol. However, it has been not reported to generate path-encoded Bell states in glass chip by femtosecond laser direct writing. In addition, we have improved the writing technique to successfully fabricate high performance chips and get a very high fidelity of ~98.8±0.3%, which provides the possibility for fabricating more complex circuits with high fidelity using this combination of logic gates or some other similar structures.
  2. In the original manuscript, we really mainly compared the silica-on-silicon platform with the glass writing by femtosecond laser without mention other platforms. We have rewritten the sentences in revised manuscript following your suggestions.

    In Introduction, we change “There are mainly two techniques to fabricate integrated photonic quantum chips, one is silicon-based lithography[1-3], and the other is the femtosecond laser direct writing (FLDW)[4-6]. Silicon-based waveguide photonic chips have maturely developed, while their waveguides…” into “They are fabricated by the silicon-based lithography[1-6], the femtosecond laser direct writing (FLDW)[7-9] , and a new emerging platform based on lithium niobate on insulator (LNOI)[10-12]. Silicon-based waveguide photonic chips have maturely developed using silica on silicon [1], silicon on insulator[4], silicon nitride[5], silicon oxynitride[6] and so on, while their waveguides…”.

   New references 4,5,6 and 10,11,12 were added.